# Effect of Lactic Acid Bacteria on Bacterial Community Structure and Characteristics of Sugarcane Juice

**DOI:** 10.3390/foods11193134

**Published:** 2022-10-08

**Authors:** Huahua Qiao, Liwei Chen, Jinsong Yang, Wenbo Zhi, Rong Chen, Tianyu Lu, Haisheng Tan, Zhanwu Sheng

**Affiliations:** 1College of Food Science and Engineering, Hainan University, Haikou 570100, China; 2College of Materials Science and Engineering, Hainan University, Haikou 570100, China; 3Haikou Experimental Station, Chinese Academy of Tropical Agricultural Sciences, Haikou 570102, China

**Keywords:** sugarcane juice, fermentation, high-throughput sequencing, high-performance liquid chromatography

## Abstract

Fermentation technology is of great significance for food preservation; through fermentation, while extending the shelf life of food, it can also improve the sensory quality of food and increase its nutritional value. Compared with natural fermentation, the use of specific microorganisms for fermentation can be used to determine the direction of fermentation. Therefore, in order to better explore the effect of bacterial community structure on the quality of sugarcane juice naturally fermented and inoculated with different lactic acid bacteria in the fermentation process, the most suitable method for sugarcane juice fermentation was found, which provided a theoretical basis for the safe production of fermented sugarcane juice. In this experiment, the sugarcane juice was treated differently and divided into four groups: natural fermentation, fermentation with *Lactobacillus* HNK10, fermentation with *Lactococcus* HNK21, and HNK10 + HNK21 compound fermentation. The changes in bacterial community structure of different treatments of sugarcane juice during fermentation were analyzed by high-throughput sequencing technology, and the quality change of different treatments of sugarcane juice during fermentation was analyzed by high-performance liquid chromatography, and the change in its bacteriostatic ability was explored. The results showed that after the sugarcane juice treated with *Lactobacillus* HNK10 was fermented at 37 °C for 48 h, the final fermented sugarcane juice had a large abundance of lactic acid bacteria and high-quality and strong antibacterial activity. Conclusions: changes in the bacterial community structure during the fermentation of sugarcane juice affect the formation of organic acids and the change of bacteriostatic ability and directly determine the quality and shelf life of fermented sugarcane juice.

## 1. Introduction

As a refreshing sugary soft drink, sugarcane juice is an important commodity in the world and an important natural health drink in subtropical and tropical countries. Sugarcane juice has suitable nutritional and organoleptic properties. Sugarcane contains a variety of vitamins, fats, proteins, organic acids, and other substances that are very beneficial to human metabolism and contain a large number of essential trace elements. Chinese medicine often uses sugarcane for medicinal use, moisturizing lung dryness, detoxification, antipyretic asthma, laxative, and other effects; therefore, sugarcane has excellent pharmacology and dietary therapeutic effects [1]. Sugarcane contains a variety of iron, calcium, phosphorus, manganese, zinc, and other essential trace elements, of which the content of iron is particularly large, up to 9 mg/kg, ranking first in the fruit, which can alleviate anemia of iron deficiency, so sugarcane has the reputation of “blood-tonifying fruit”. Studies have found that sugarcane juice contains a variety of physiologically active substances, such as flavonoids and polyphenols. These active substances can scavenge free radicals in the body, thereby playing an antioxidant effect [2]. However, the sugar content of sugarcane juice is very high, with a sugar content of up to 17–18%. The bacteria and yeasts will cause the sugarcane juice to ferment rapidly in the juice, leading to changes in its color, taste, and flavor [3]. It is particularly prone to spoilage; the production of toxic substances may cause nerve damage, dizziness, headache, nausea, and other uncomfortable symptoms, posing a threat to the health of consumers, which is also the main obstacle to the industrial-scale production of sugarcane juice [4].

Biological preservation refers to the use of specific microorganisms and their metabolites to extend shelf life and improve food safety. Compared to chemical and physical preservation methods, biological preservation is more easily accepted by the majority of consumers, lactic acid bacteria (LAB) fermentation is a common biological preservation method, and LAB antibacterial activity is mainly based on the production of lactic acid, organic acids, hydroperoxides and bacteriocins, and other metabolites [5]. Through LAB fermentation, not only can the shelf life be extended, but also the sensory quality of food can be improved, and a large number of studies have shown that lactic acid fermentation has a positive effect on nutritional value and improved digestibility of raw materials after fermentation will produce a large number of lactic acid, amino acids, short-chain fatty acids, and other nutrients and probiotic metabolites, these products can regulate the gastrointestinal flora, reduce the activity of cholesterol and play an immunomodulatory role. After fermentation, the juice of fruits and vegetables is not only rich in vitamins and minerals but also has many active LABs, which is conducive to the improvement in the intestinal environment. The health-promoting properties of LAB are mainly based on increased bioavailability of nutrients, antioxidant activity, biosynthesis of vitamins, and degradation of anti-nutrients [6]. The composition of the media flora in the fermentation samples of LAB is complex, and its variety, quantity, and range are large. The addition of specific LAB for fermentation changes the relevance of the microbial community in the sample and the dominant genus, which is closely related to the quality of the fermented sample [7].

Many researchers have studied various preservation methods of sugarcane juice. For example, the combination of natural preservatives and low-temperature storage to treat sugarcane juice is an effective preservation method that can be stored for more than one month and has satisfactory organoleptic qualities [8]. By adding LAB to sugarcane juice for fermentation, the fermented sugarcane juice drink contains many minerals, showing suitable antioxidant capacity and suitable sensory evaluation. Therefore, by using LAB to ferment sugarcane juice while improving its nutritional value and obtaining suitable sensory quality, it is feasible to extend the shelf life of sugarcane juice, and at the same time, the biological preservative based on LAB and their metabolites also has certain safety and is more easily accepted by consumers.

Therefore, *Lactobacillus* HNK10 and *Lactococcus* HNK21 with fast acid production, fast growth, and high-temperature resistance were selected from the strains preserved in the laboratory and added to the fresh sugarcane juice for sugarcane juice fermentation test. The aims of this study are to explore the effect of LAB on the bacterial community structure and characteristics of sugarcane juice, find out the most suitable method for sugarcane juice fermentation, and provide theoretical knowledge for the large-scale production of sugarcane juice.

## 2. Materials and Methods

### 2.1. Materials and Main Equipment

Fresh sugarcane was commercially available in Haikou. Vitamin C powder (food grade), NA, LB, and MRS medium base were purchased from Guangdong Huankai Microbial SCI. & Tech. Co., Ltd. (Guangzhou, China). An aqueous solution of perchloric acid and methanol (Shanghai, China) was prepared for ultra-performance liquid chromatography (UPLC) analysis. Organic acid standards (L-lactic acid, malic acid, oxalic acid, and galacturonic acid) were obtained from Shanghai Aladdin Biochemical Technology Co., Ltd. (Shanghai, China). HNK10 (*Lactobacillus*) and HNK21 (*Lactococcus*)were obtained from the laboratory of the College of Food Science and Engineering, Hainan University. Laboratory PH meters were obtained from OHAUS Instruments Ltd. (Changzhou, China). High-performance liquid chromatograph Agilent1260, Agilent. (Santa Clara, CA, USA).

### 2.2. Seed Liquid Preparation

(*Lactobacillus*) HNK10 and (*Lactococcus*) HNK21 were respectively inoculated into MRS medium and cultured at 37 °C for 24 h. Then, it was transferred to a sugarcane juice culture medium and cultured at 37 °C for 48 h [9].

### 2.3. Sample Preparation of Fermented Sugarcane Juice

Fresh sugarcane was washed and juiced, adding 0.05% VC. We divided 150 mL into 50 mL volumetric flasks after sterilization for fermentation of fresh sugarcane juice. Another 1000 mL was pasteurized in a water bath at 80 °C for 25 min and then divided into 50 mL volumetric flasks after sterilization. Each LAB seed solution was respectively inoculated into the sterilized sugarcane juice at an inoculum of 5% per 50 mL. The sample was fermented at 37 °C. In this experiment, the sugarcane juice was treated differently and divided into four groups: natural fermentation, fermentation with HNK10, fermentation with HNK21, and compound fermentation with HNK10 + HNK21. Four groups of fermentation broth were sampled at 0 h, 12 h, 24 h, 36 h, and 48 for the fermentation time, and the changes in bacterial community structure of different processed sugarcane juice during fermentation were analyzed by high-throughput sequencing technology. Entrusted Biomark Biotechnology Co., Ltd. (Shandong, China) to perform high-throughput sequencing.

### 2.4. Determination of Organic Acid Content in Fermented Sugarcane Juice

#### 2.4.1. Sample Preparation

The sample was centrifuged at 10,000 r/min at 4 °C for 5 min, and the supernatant was taken for the determination of organic acid content.

#### 2.4.2. Chromatographic Conditions

Column: Agilent Zorbax SB-aq (250 mm × 4.6 mm, 5 m); column temperature 25 °C; injection volume 10 µL; flow rate 1.0 m L/min; mobile phase A was methanol (80%), B was an aqueous solution of perchloric acid (0.3%, pH 2.4), and the volume ratio was A:B = 2:98; the same amount is eluted for 15 min, and the detection wavelength is 210 nm.

### 2.5. Determination of Antibacterial Ability of Fermented Sugarcane Juice

The fermentation broth was taken every 12 h and centrifuged (4000 r/min, 4 °C, 5 min), and the supernatant was collected after centrifugation. Using the coating plate method, *Salmonella*, *Escherichia coli*, and *Staphylococcus aureus* were evenly spread into NA and LB medium, respectively. The sterilized filter paper was soaked in the supernatant of the fermentation broth for 30 min. Drain the soaked filter paper and spread it on each flat plate, and use the unsoaked filter paper as a blank control in the middle. After culturing in a constant temperature incubator at 37 °C for 24 h, record whether there is an inhibition zone around the filter paper and determine the diameter of the inhibition zone [10].

### 2.6. Sensory Analysis

The appearance, color, taste, and aroma of fermented sugarcane juice under different conditions were evaluated based on a simple sensory analysis. A score from 0 to 5 was awarded to each sample by 10 experienced members of the laboratory, with 0 being the “least liked” for each participant and 5 being the “most liked” [11].

### 2.7. Statistical Analysis

The data were analyzed by SPSS 18.0 software; image processing used Origin (Version 8.6); high-throughput sequencing results were optimized by software such as FLASH (v1.2.11).

## 3. Results and Discussion

### 3.1. Sequencing Results of Fermented Sugarcane Juice

The original sequence data of microorganisms in fermented sugarcane juice were obtained by high-throughput sequencing, and the data were optimized by FLASH. A total of 243,330 effective bacterial sequences were obtained. The average number of operational taxonomic units (OTU) that can be used for species classification was 185, covering at least 16 phyla, 23 classes, 40 orders, 67 families, 121 genera, and 133 species. The statistical map of bacterial high-quality sequence distribution was finally obtained by data preprocessing (Figure 1). The sequencing length is mainly concentrated in the interval length of 370–440 bp, which is similar to the amplification length of the designed primers.

### 3.2. Alpha Diversity Analysis

#### 3.2.1. Dilution Curve

The dilution curve is used to evaluate the coverage of all groups in the tested sample in the high-throughput sequencing results, which can indirectly reflect the abundance of species in the sample [12]. When the curve flattens, it means that the sequencing depth is large enough and the species diversity in the sample is covered. On the contrary, it means that the sequencing depth is not enough, and the species in the sample are not completely covered.

Figure 2 is a sample dilution curve, from which it can be seen that as the number of sample sequences increases, the number of OTU in each sample increases, and when the number of OTU reaches a certain number, the curve tends to flatten, indicating that in the sequencing process of the sugarcane juice sample, the sequencing depth is sufficient, and the vast majority of bacterial community information in the sugarcane juice sample has been covered. If you expand the sequencing depth again, there is no obvious significance. In addition, it can be clearly seen from Figure 2 that the sugarcane fermentation juice treated with HNK10 has the highest species richness, and the natural fermentation of fresh sugarcane juice has the lowest species richness.

#### 3.2.2. Veen Diagram

Venn diagram reflects the similarity and difference of microbial communities among different samples. It can be seen from Figure 3 that there are 182 OTU that are shared by the sugarcane juice of different treatments, there are 2 OTU common to three treatment groups, 1 OTU common to sugarcane juice with HNK10 added and sugarcane juice with HNK10 + HNK21 added, and 0 OTU unique to each sample. Therefore, it can be seen that there are significant differences in microbial communities among the samples.

#### 3.2.3. Alpha Diversity Index Analysis

The Alpha Diversity index of fermented sugarcane juice microorganisms is shown in Table 1. Alpha diversity refers to the diversity in the tested sample system. Chao 1 and Ace are usually used to express the abundance of the flora, Shannon index and Simpson diversity index are used to express the diversity of the flora, and the identification level is 97% [13].

It can be seen from Table 1 that the coverage ratios are all greater than 0.997. The coverage results indicate that the sequence in the sugarcane juice is almost completely detected, and the sequencing results represent the true situation of the bacterial community in the sugarcane juice. With the increase in fermentation time, the Chao1, Ace, Shannon index, and Simpson index in sugarcane juice added with HNK10 or HNK21 or compound treatment showed a trend of first decreasing, then increasing, and finally decreasing. In the control group, the indexes of fresh sugarcane juice fermentation showed there is a tendency to increase first and then decrease. It shows that with the progress of fermentation, the abundance and diversity of bacterial communities in each group have changed.

### 3.3. Species Composition Analysis

#### 3.3.1. Analysis of Flora Structure of Fermented Sugarcane Juice Based on Phylum Level

As shown in Figure 4, at the phylum classification level, a total of 11 bacterial phyla were detected in fermentation samples, mainly including *Firmicutes*, *Proteobacteria*, *Cyanobacteria* and others. *Firmicutes* is the absolute dominant phylum of the treatment group (addition of HNK10 sugarcane juice, the addition of HNK21 sugarcane juice, compound addition of sugarcane juice), reaching more than 60%. *Proteobacteria* accounted for a higher proportion (46%) in the fermented fresh sugarcane juice of the control group. The abundance of *Bacteroidetes* and *Cyanobacteria* is very low, but they always exist during the fermentation process.

As can be seen from Figure 4, in the natural fermentation group, the *Proteobacteria* is the absolute dominant phylum, the *Firmicutes* is the absolute dominant phylum of the other three treatment groups, and the control group has a significant difference from the treatment group at the phylum level, and this difference may be caused by natural fermentation. Because the control group used natural fermentation, while the treatment group used the addition of LAB. The research showed that under different processing conditions, the addition of LAB will change the correlation between the microbial flora and the dominant bacteria in the fermentation sample [14].

#### 3.3.2. Analysis of the Microbial Community Structure of Fermented Sugarcane Juice Based on the Genus Level

At the genus classification level, as shown in Figure 5. A total of 11 bacterial genera were detected in fermented sugarcane juice with different treatments, and those with a relative abundance of less than 1% were combined into one category (other). It can be seen from the figure that *Lactobacillus* is the absolute dominant genus of the treatment group. The proportions in sugarcane juice with added HNK10 and added HNK21, and combined treatments were 56%, 72%, and 58%, respectively. *Lactobacillus*, *Pantoea*, and *Leuconostoc* are the dominant genera in the control group.

### 3.4. Significant Difference Analysis between Groups

LefSe analysis is the analysis of species with significant differences between groups to determine whether there are significantly different species in different groups [15]. It can be seen from Figure 6a that the control group (fresh sugarcane juice) fermentation and other treatment groups have significant differences in the abundance of *Lactobacillus*. The control group had the smallest abundance, and the fermented sugarcane juice treated with HNK10 had the greatest abundance. It can be seen from Figure 6b that there is a significant difference in the abundance of *Leuconostoc* between the fermentation control group and other treatment groups. The control group had the greatest abundance of *Leuconostoc*, and the remaining groups were basically non-existent *Leuconostoc*. It can be seen from Figure 6c that there is a significant difference in the abundance of *Enterobacter* between the fermentation control group and other treatment groups. The abundance of *Enterobacter* in the control group was the greatest, and the abundance of *Enterobacter* in fermented sugarcane juice treated with HNK10 was the smallest.

The natural fermentation of sugarcane juice is a spontaneous process, the fresh sugarcane juice has not undergone any sterilization treatment, so there are a large number of *Pantoea* and *Enterobacter* during the fermentation process, which affects the quality of fermented sugarcane juice. In addition, the abundance of LAB will also affect the fermentation quality. If lactic acid fermentation cannot effectively reduce the pH or inhibit unfavorable bacteria, insufficient epiphytic lactic acid bacteria may also lead to poor fermentation quality [16]. As can be seen from Figure 6a–c, the control group and other treatment groups have obvious differences in the abundance of *Lactobacillus*, *Leuconostoc,* and Enterobacter, and it can be seen from the structural analysis of the sugarcane juice flora that the *Lactobacillus* is the dominant bacteria in the control group and the treatment group, *Leuconostoc* and *Enterobacter* are the dominant bacteria in the control group, the greater the abundance of the *Lactobacillus* and the *Leuconostoc*, the higher the quality of the fermentation broth, and the smaller the abundance of *Enterobacter*, indicating that the quality of the fermentation broth is better, therefore, From the analysis of Figure 6, it can be seen that the fermented sugarcane juice with HNK10 treatment is of the best quality.

### 3.5. Similarity Cluster Analysis

After standardization based on the OTU data, the top 80 species with the largest number were selected and mapped based on the R heat map. Each color block in the heat map represents the abundance of a genus in a sample. In the heat map, by arranging the samples horizontally and the species vertically, the similarity between the samples and the similarity of the community composition at each classification level can be understood in the cluster map [17]. It can be seen from Figure 7 that the community composition of fermented sugarcane juice added with HNK10, HNK21, or combined treatment is the closest, while the natural fermented fresh sugarcane juice of the control group is different from other samples.

### 3.6. Principal Component Analysis

#### 3.6.1. PCA

It can be seen from Figure 8 that different treatment methods have an impact on the composition of the bacterial flora of fermented sugarcane juice. The contribution rate of bacterial principal component 1 (PC1) and principal component 2 (PC2) to the difference of sugarcane juice samples reached 45.64% and 23.7%, respectively, and a total of 79.34% of the variables can be explained. In the PCA analysis, the greater the distance between each point, the greater the difference in flora between them [18]. The fresh sugarcane juice and sugarcane juice supplemented with HNK10 in the control group are far away from the other two groups, indicating that the flora structure is very different between them. The sugarcane juice fermented with HNK21 and the sugarcane juice fermented by the compound fermentation is closer, indicating that the flora of the two groups are similar in structure.

#### 3.6.2. Non-Metric Multidimensional Scaling

The non-metric multidimensional scaling method, or NMDS, is a way that simplifies samples from multidimensional space to low-dimensional space for analysis and classification. The main difference between NMDS and PCA analysis lies in the evolutionary information [19]. It can be seen from Figure 9 that the results are consistent with the PCA analysis results, which further proves the validity of the PCA results.

### 3.7. Changes in pH and Organic Acids in Sugarcane Juice Fermentation

The changes in sugarcane juice quality are shown in Table 2. As time went on, the pH in the fresh sugarcane juice and the sugarcane juice treated with different lactic acid bacteria showed a downward trend. Except for 36 h, there were significant differences in each group. In addition, in order to evaluate the changes in the concentration of organic acids during the fermentation process. The organic acids in sugarcane juice were analyzed by high-performance liquid chromatography. Organic acid is the key to making up the flavor substances in fermented products and measuring the degree of fermentation. The content of organic acid is directly related to the quality of the final fermented product [20].

During the fermentation process, the lactic acid concentration of the control group and the HNK21-treated group showed a trend of first increasing and then decreasing. The lactic acid concentration of the *Lactobacillus* HNK10 and *Lactococcus* HNK21 combined treatment group increased with the increase in fermentation time. After 24 h of fermentation, there were significant differences among the groups. At 24 h, the lactic acid content of the control group and HNK21-treated group reached the maximum; at 48 h, the lactic acid content of the HNK10 treatment group and the combined treatment group reached the maximum.

Compared with other organic acids produced during the fermentation of sugarcane juice, malic acid accounts for the largest proportion of fermented sugarcane juice. The sourness of malic acid is 1.2 times that of citric acid, and its sourness remains in the mouth for a long time. It is outstanding in the effect of strengthening the sourness, which can improve the over-sweet flavor of sugarcane juice. In addition, malic acid is highly safe and is often used to prepare beverages. During the fermentation of sugarcane juice, the concentration of malic acid in each group continued to increase, and the difference was significant.

During the fermentation process, the oxalic acid and galacturonic acid of the treatment group and the control group also showed a trend of increasing first and then decreasing. The oxalic acid was significantly different at 48 h, while the galacturonic acid was significantly different at 36 h.

The pH of the sugarcane juice treatment group decreased with the increase in fermentation time, while the pH of the control group decreased first and then increased. The decrease in pH in each group is mainly due to the conversion of carbohydrates by LAB into organic acids (such as lactic acid and malic acid). The rapid decrease in pH helps to inhibit the activities of undesirable microorganisms in the early stage of sugarcane juice fermentation, reduce the risk of raw material spoilage, and preserve nutrients. At 48 h in the late fermentation stage, the pH of the fresh sugarcane juice in the control group began to rise and was higher than that of the treatment group. According to the high-throughput sequencing results, this was due to the insufficient abundance of *Leuconostoc* in the fresh sugarcane juice leading to the growth of undesirable microorganisms. The pH of the LAB treatment group was basically stable at 48 h. The pH value of sugarcane juice fermented by *Lactobacillus* HNK10 was the lowest at 3.10, which was similar to that of sugarcane juice co-fermented with *Lactobacillus* and *Lactobacillus*, which was better than the fermentation result of *Lactococcus* HNK21. It is also consistent with the results of LefSe analysis in high-throughput sequencing.

Studies have shown that the metabolites produced by the fermentation of different lactic acid bacteria can cause chemical and physical changes in the fermented juice. Due to the interaction between enzyme activity, substrate components, and metabolites produced by fermenting microorganisms, the fermentation process can improve the nutritional value and flavor of the fermentation substrate [21]. The content of lactic acid in fruits and vegetables is small, it has an antiseptic effect, and the acidity is weaker than malic acid. It can adjust the pH so that the fermented sugarcane juice has a strong lactic acid flavor and prevents the growth of miscellaneous bacteria. During the fermentation process, the lactic acid concentration of the control group and the lactic acid bacteria alone treatment group showed a trend of first increasing and then decreasing. There was no significant difference among the groups, and the concentration was the highest at 12 h in the early stage of fermentation. *Lactobacillus* HNK10 and *Lactococcus* HNK21 combined treatment group lactic acid concentration increased with the increase in fermentation time, reaching the maximum at 48 h. Combined with the high-throughput sequencing results, it can be seen that the main reason is that the composite lactic acid bacteria treatment group and the HNK10 treatment group have a sufficient abundance of LAB to produce a large amount of lactic acid.

Compared with the organic acids produced during the fermentation of sugarcane juice, malic acid accounts for the largest proportion of fermented sugarcane juice. During the fermentation of sugarcane juice, the concentration of malic acid in each group continued to increase.

During the fermentation process, the oxalic acid and galacturonic acid of the treatment group and the control group also showed a trend of increasing first and then decreasing, and the difference was not significant. Oxalic acid is widely found in plant-derived foods and is a strong acid among organic acids. Galacturonic acid is a constituent unit of pectic acid and an important component in pectin. Although pectin is part of the human diet, it has no significant contribution to nutrition. The identification of the main microorganisms related to organic acids requires further analysis. These organic acids may have an impact on the final flavor of fermentation [22].

### 3.8. Changes in Bacteriostatic Ability during Fermentation of Sugarcane Juice

As can be seen from Table 3, fresh sugarcane juice has a certain inhibitory effect on *Salmonella*, and when it is not fermented, the diameter of the bacteriostatic circle is more than 11 mm, and the bacteriostatic effect is obvious. After fermentation, sugarcane juice showed obvious antibacterial activity, and there were significant differences in inhibition of *Salmonella* and *E. coli* in the control group and HNK21 treatment group, the HNK10 treatment group, and the compound treatment group. *Lactobacillus* HNK10 has a stronger inhibitory effect on *Salmonella* than *Lactococcus* HNK21, and *Lactococcus* HNK21 has a strong inhibitory ability on *Escherichia coli*. The combined fermentation of sugarcane juice from HNK10 + HNK21 has the best inhibition effect on *Salmonella*, *E. coli* and *Staphylococcus aureus*.

Many studies have reported that LAB controls microbial growth by producing acids and or antimicrobial substances [23]. The increase in the antibacterial capacity of fruit juice during fermentation has also been reported in a large number of previous studies [24]. The findings showed that adding *Lactobacillus* to sweet lemon juice for fermentation can significantly improve the antibacterial effect of the juice [25]. The results of this study were consistent with those of the general study, as the antibacterial activity of the control and treatment groups increased after 48 h of fermentation. From the high-throughput results of fermented sugarcane juice, it can be seen that the control group produced *Leuconostoc* during the fermentation process, and *Leuconostoc* had bacteriostatic effect. A large number of LAB were present in the treatment groups, and LAB were able to produce various antimicrobial compounds such as lactic acid, propionic acid, etc., which can reduce the pH value and inhibit many microorganisms. In addition, LAB provide a wide range of bacteriostatic substance [26]. Studies have shown that LAB plays an active role in displaying antimicrobial activity against common foodborne pathogens [27]. The treatment group showed excellent bacteriostatic ability after fermentation.

### 3.9. Sensory Analysis

The samples were ranked from worst to best in the following order: fresh sugarcane juice, fresh sugarcane juice + HNK21, fresh sugarcane juice + HNK10 + HNK21, and fresh sugarcane juice + HNK10. Fresh sugarcane juice is naturally fermented at 37 °C for 48 h, with poor taste. However, the sugarcane juice fermented with LAB is sweet and sour and has an excellent drinking effect. Its shelf life is 8 months under a sealed environment.

The outcome of the sensory analysis is shown in Table 4.

## 4. Conclusions

In this study, high-throughput sequencing and high-performance liquid chromatography were used to analyze the bacterial community structure and quality changes of naturally fermented sugarcane juice and sugarcane juice fermented with different lactic acid bacteria. The results show that different fermentation methods will affect the bacterial community structure and quality of fermented sugarcane juice. There are LABs, such as *Leuconostoc*, in the fresh sugarcane juice that can undergo natural fermentation and also exhibit a certain antibacterial ability. However, the fermentation quality is not as suitable as the sugarcane juice treated with lactic acid bacteria. Compared with HNK21 and the composite group, sugarcane juice treated with *Lactobacillus* HNK10 was fermented at 37 °C for 48 h, and the final fermented sugarcane juice had a large abundance of LAB, high lactic acid content, and strong antibacterial activity. It shows that adding *Lactobacillus* HNK10 to sugarcane juice can improve the fermentation quality of sugarcane juice, which has practical guiding significance for the sugarcane juice fermentation industry.

## Figures and Tables

**Figure 1 foods-11-03134-f001:**
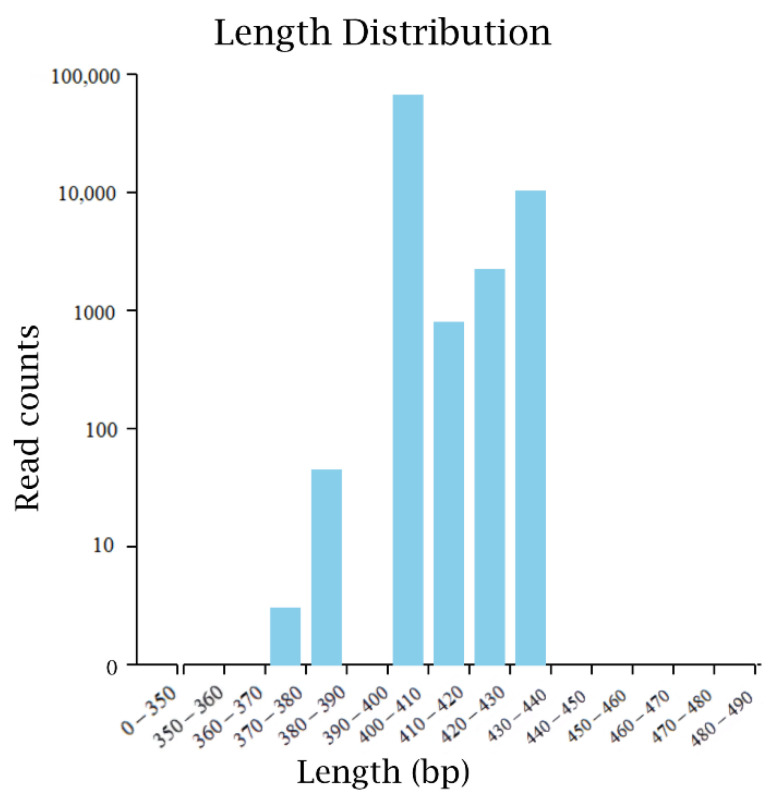
The high-quality sequence distribution of bacteria in the high-through sequencing.

**Figure 2 foods-11-03134-f002:**
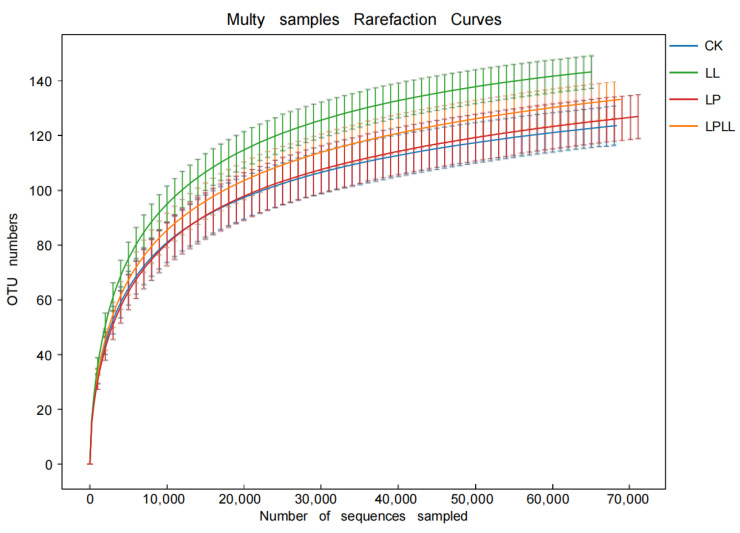
OTU curve of fermented sugarcane juice. CK: fresh sugarcane juice; LL: sugarcane juice with added HNK10; LP: sugarcane juice added with HNK21; LPLL: sugarcane juice treated with HNK10 + HNK21 was added.

**Figure 3 foods-11-03134-f003:**
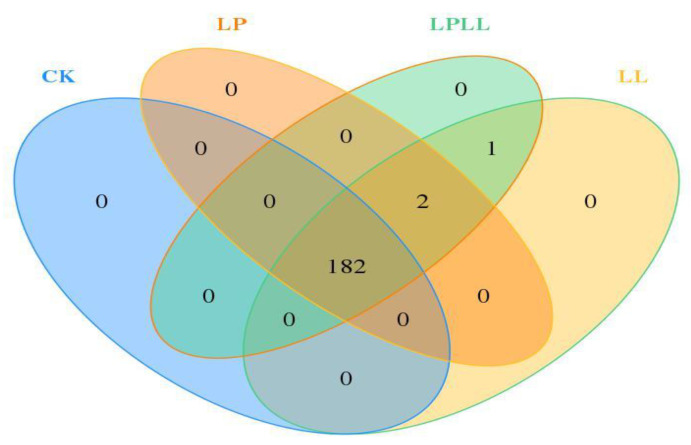
Veen Diagram of fermented sugarcane juice. CK: fresh sugarcane juice; LL: sugarcane juice with added HNK10; LP: sugarcane juice added with HNK21; LPLL: sugarcane juice treated with HNK10 + HNK21 was added.

**Figure 4 foods-11-03134-f004:**
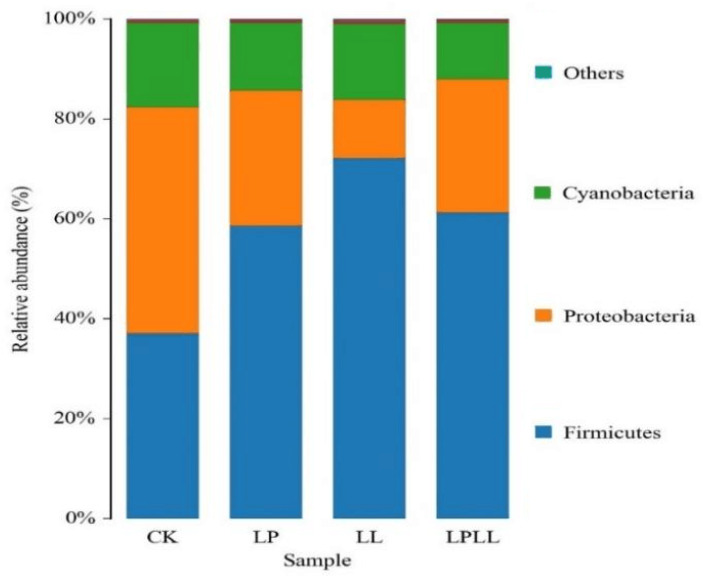
UPGMA cluster analysis of bacteria relative abundance on phylum level. CK: fresh sugarcane juice; LL: sugarcane juice with added HNK10; LP: sugarcane juice added with HNK21; LPLL: sugarcane juice treated with HNK10 + HNK21 was added.

**Figure 5 foods-11-03134-f005:**
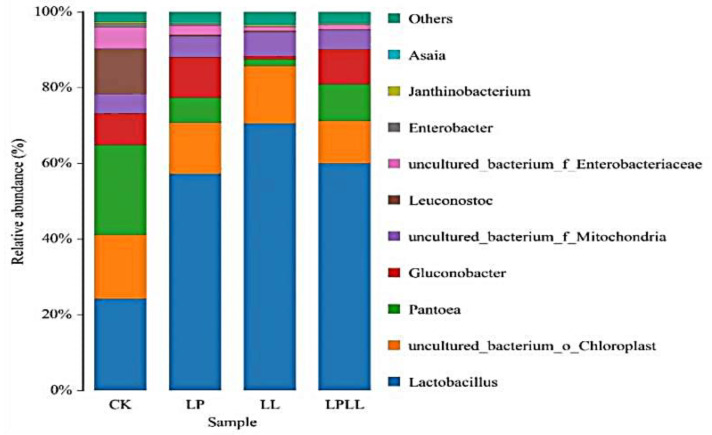
UPGMA cluster analysis of bacteria relative abundance on Genus level. CK: fresh sugarcane juice; LL: sugarcane juice with added HNK10; LP: sugarcane juice added with HNK21; LPLL: sugarcane juice treated with HNK10 + HNK21 was added.

**Figure 6 foods-11-03134-f006:**
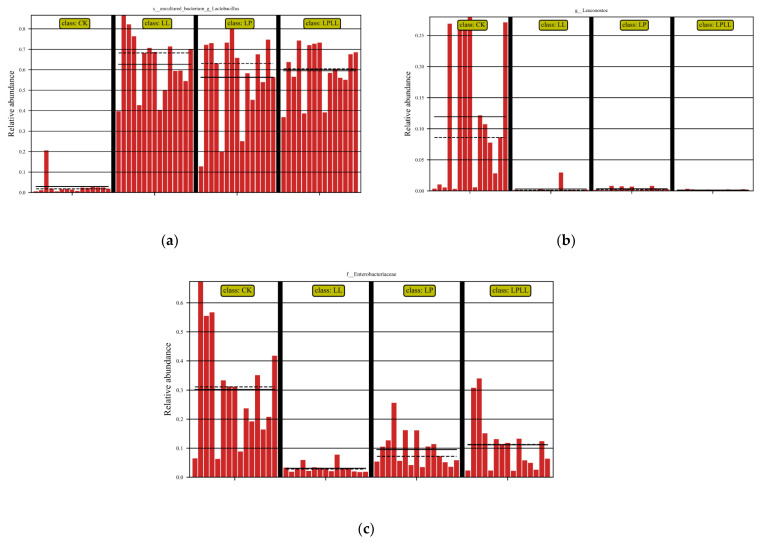
LefSe analysis of fermented sugarcane juice. (**a**): abundance of *Lactobacillus*. (**b**): abundance of *Leuconostoc*. (**c**): abundance of *Enterobacter*. CK: fresh sugarcane juice; LL: sugarcane juice with added HNK10; LP: sugarcane juice added with HNK21; LPLL: sugarcane juice treated with HNK10 + HNK21 was added.

**Figure 7 foods-11-03134-f007:**
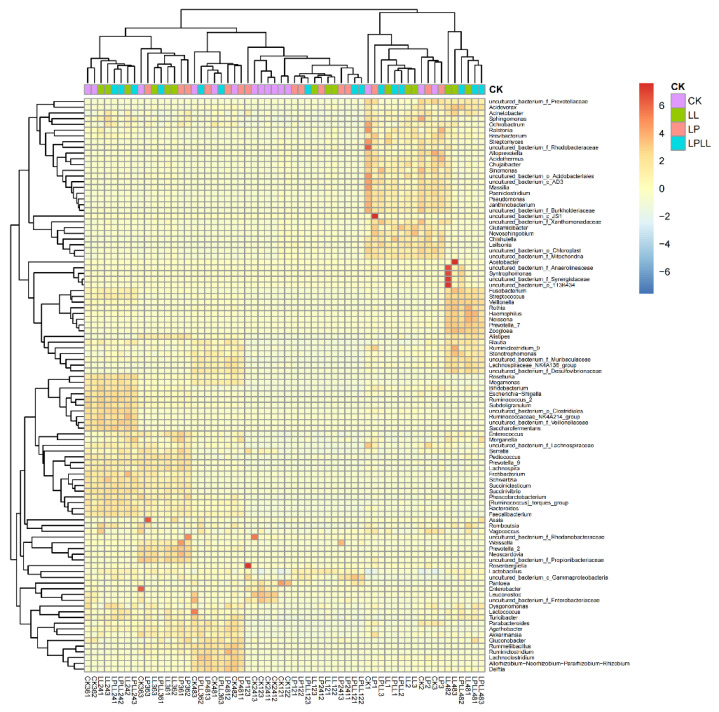
R heat map of fermented sugarcane juice.CK: fresh sugarcane juice; LL: sugarcane juice with added HNK10; LP: sugarcane juice added with HNK21; LPLL: sugarcane juice treated with HNK10 + HNK21 was added.

**Figure 8 foods-11-03134-f008:**
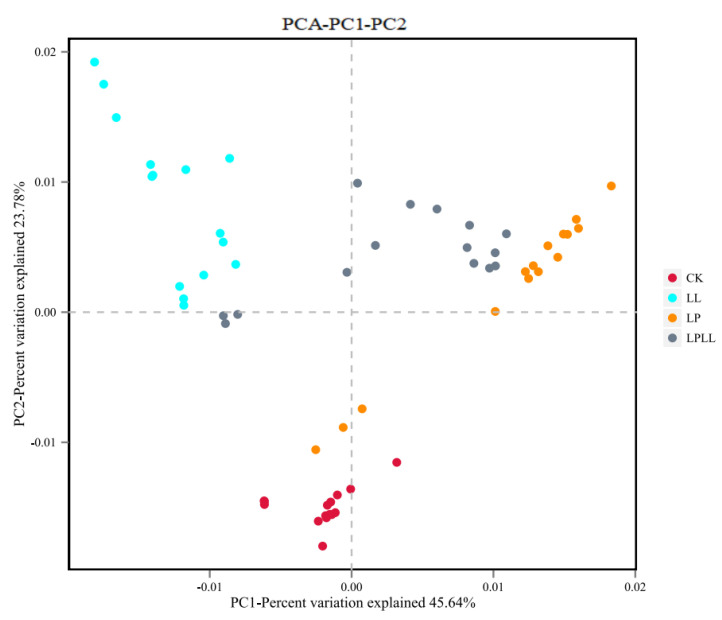
PCA of fermented sugarcane juice. CK: fresh sugarcane juice; LL: sugarcane juice with added HNK10; LP: sugarcane juice added with HNK21; LPLL: sugarcane juice treated with HNK10 + HNK21 was added.

**Figure 9 foods-11-03134-f009:**
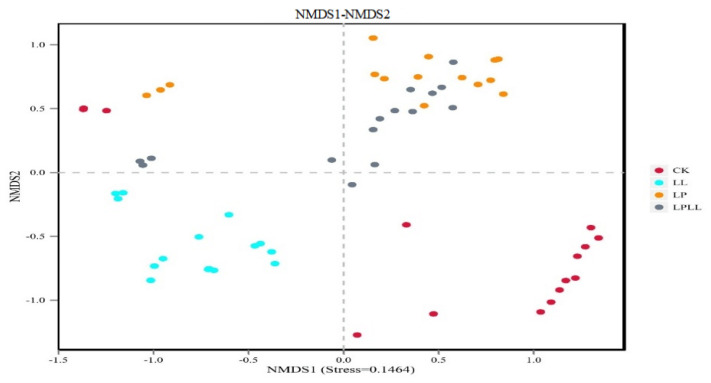
NMDS of fermented sugarcane juice. CK: fresh sugarcane juice; LL: sugarcane juice with added HNK10; LP: sugarcane juice added with HNK21; LPLL: sugarcane juice treated with HNK10 + HNK21 was added.

**Table 1 foods-11-03134-t001:** Diversity index of microorganisms in fermented sugar cane juice.

Fermentation Method	Fermentation Time	Shannon	Simpson	ACE	Chao 1	Coverage
Fresh sugarcane juice	0 h	1.9394	0.5361	146.6979	146.0889	0.9999
	12 h	1.9963	0.6249	147.1516	147.1516	0.9997
	24 h	2.5556	0.7778	149.5131	148.9484	0.9998
	36 h	2.7124	0.7456	156.9436	154.4762	0.9999
	48 h	2.6063	0.7422	152.1198	153.7333	0.9999
Fresh sugarcane juice + HNK10	0 h	2.0431	0.6712	146.9601	151.4889	0.9999
	12 h	1.1348	0.3352	123.1293	117.1402	0.9999
	24 h	2.0618	0.5220	171.0971	173.7381	0.9998
	36 h	2.4813	0.6830	173.4629	174.6033	0.9999
	48 h	2.4406	0.6662	165.8255	165.9692	0.9999
Fresh sugarcane juice + HNK21	0 h	2.3088	0.6744	154.4233	155.7787	0.9999
	12 h	1.6568	0.5055	137.4734	124.3867	0.9997
	24 h	1.478	0.4279	139.1534	125.3903	0.9999
	36 h	2.2562	0.6101	168.8637	169.4510	0.9999
	48 h	1.8172	0.5355	162.1182	164.6531	0.9999
Fresh sugarcane juice + HNK10 + HNK21	0 h	2.3131	0.7228	145.1451	144.6062	0.9997
	12 h	1.1980	0.4131	136.0097	117.2084	0.9999
	24 h	2.5116	0.7772	170.7681	171.4101	0.9999
	36 h	2.5926	0.7726	179.3964	172.0490	0.9999
	48 h	2.3103	0.6430	153.6262	160.1191	0.9999

**Table 2 foods-11-03134-t002:** Changes in pH and organic acids during sugarcane juice fermentation.

	Time/h	Fresh Sugarcane Juice	Fresh Sugarcane Juice + HNK10	Fresh Sugarcane Juice + HNK21	Fresh Sugarcane Juice + HNK10 + HNK21
pH	0	4.82 ± 0.01 a	4.76 ± 0.01 b	4.46 ± 0.04 c	4.83 ± 0.00 a
	12	4.51 ± 0.02 a	3.86 ± 0.01 b	3.75 ± 0.01 c	3.56 ± 0.01 d
	24	3.58 ± 0.02 a	3.37 ± 0.01 b	3.48 ± 0.01 c	3.35 ± 0.00 d
	36	3.23 ± 0.01 a	3.15 ± 0.02 a	3.25 ± 0.28 a	3.15 ± 0.01 a
	48	3.4 ± 0.00 a	3.10 ± 0.01 bd	3.25 ± 0.00 cd	3.13 ± 0.00 bcd
Lactic acid (mg/mL)	0	0 ± 0.00 a	0.06 ± 0.00 a	0.09 ± 0.00 a	0.07 ± 0.00 a
	12	0.57 ± 0.02 ab	0.36 ± 0.01 c	0.45 ± 0.01 b	0.55 ± 0.01 a
	24	0.61 ± 0.01 a	0.47 ± 0.01 b	0.61 ± 0.07 a	0.57 ± 0.03 a
	36	0.51 ± 0.02 ab	0.53 ± 0.00 ab	0.60 ± 0.00 c	0.63 ± 0.00 d
	48	0.11 ± 0.01 a	0.54 ± 0.02 b	0.27 ± 0.02 c	0.71 ± 0.01 d
Malic acid (mg/mL)	0	2.9 ± 0.01 a	2.48 ± 0.01 b	2.48 ± 0.03 b	2.79 ± 0.01 c
	12	4.15 ± 0.06 a	5.31 ± 0.02 b	3.39 ± 0.21 c	5.39 ± 0.11 b
	24	5.47 ± 0.13 a	6.78 ± 0.00 b	4.94 ± 0.09 c	7.20 ± 0.06 d
	36	6.11 ± 0.01 a	7.84 ± 0.04 b	4.94 ± 0.00 c	8.41 ± 0.02 d
	48	7.76 ± 0.00 a	8.42 ± 0.01 b	6.48 ± 0.04 c	8.94 ± 0.07 d
Oxalic acid (mg/mL)	0	0.03 ± 0.01 a	0.06 ± 0.01 b	0.07 ± 0.01 b	0.06 ± 0.00 b
	12	0.19 ± 0.00 a	0.15 ± 0.13 a	0.17 ± 0.02 a	0.20 ± 0.02 a
	24	0.15 ± 0.01 a	0.14 ± 0.02 a	0.15 ± 0.00 a	0.20 ± 0.02 a
	36	0.12 ± 0.01 a	0.12 ± 0.01 a	0.14 ± 0.00 a	0.08 ± 0.01 a
	48	0.11 ± 0.02 a	0.03 ± 0.00 b	0.03 ± 0.01 b	0.06 ± 0.00 c
Galacturonic acid (mg/mL)	0	0.89 ± 0.01 a	0.77 ± 0.06 b	0.60 ± 0.01 c	0.94 ± 0.01 a
	12	0.91 ± 0.31 a	0.89 ± 0.01 a	0.81 ± 0.00 b	0.90 ± 0.07 a
	24	0.84 ± 0.00 a	0.89 ± 0.01 a	0.87 ± 0.05 a	0.82 ± 0.01 a
	36	0.68 ± 0.01 a	0.79 ± 0.01 b	0.83 ± 0.02 c	0.82 ± 0.00 c
	48	0.03 ± 0.00 a	0.60 ± 0.01 a	0.94 ± 0.01 a	0.73 ± 0.06 a

Note: a–d indicate significant differences between data in the same column (*p* < 0.05).

**Table 3 foods-11-03134-t003:** Bacteriostatic ability of sugarcane juice during fermentation (unit: mm).

Treatment Method	Fermentation Time	*Salmonella*	*E. coli*	*Staphylococcus aureus*
Fresh sugarcane juice	0 h	11.1 ^a^	0 ^a^	0 ^a^
	48 h	12.0 ^a^	13.1 ^a^	11.1 ^a^
Fresh sugarcane juice + HNK10	0 h	11.6 ^b^	0 ^a^	12.0 ^b^
	48 h	13.2 ^b^	14.2 ^b^	17.2 ^b^
Fresh sugarcane juice + HNK21	0 h	11.0 ^a^	0 ^a^	0 ^a^
	48 h	11.5 ^a^	16.1^c^	17.0 ^b^
Fresh sugarcane juice + Compound treatment (HNK10 + HNK21)	0 h	12.1 ^c^	0 ^a^	12.0 ^b^
	48 h	13.0 ^b^	14.9 ^d^	17.3 ^b^

Note: a–d indicate significant differences between data in the same column (*p* < 0.05).

**Table 4 foods-11-03134-t004:** Results of the sensory analysis. The average rating was ranked from least liked (0) to most liked (5).

Treatment Method	Average Rating	Comments
Fresh sugarcane juice	1	The appearance is pale yellow, the color is uneven, there is a small amount of precipitation, there is a peculiar smell, and the taste of sugarcane is too light.
Fresh sugarcane juice + HNK10	5	The appearance is yellow, the color is uniform, and there is no peculiar smell. In addition to retaining the fragrant smell of sugarcane, there is a light flavor of lactic acid, sweet, and sour.
Fresh sugarcane juice + HNK21	3	
Fresh sugarcane juice + Compound treatment (HNK10 + HNK21)	4	

## Data Availability

The data are available from the corresponding author.

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
