# Peer review of "Effect of Lactic Acid Bacteria on Bacterial Community Structure and Characteristics of Sugarcane Juice"

_foods, 2022, doi:10.3390/foods11193134_

Round 1

Reviewer 1 Report

There is absence of sensory analysis and shelf life in the study which is important that makes the study incomplete. The results of both sensory and shelf life are necessary.

1.     Title isn’t appropriate

It can be “Effect of lactic acid bacteria on bacterial community structure and characteristics of sugarcane juice”.

2.     Similarly, objectives mentioned in introduction need to be rewritten.

3.     In materials and method

(i)              Past tense should be used.

(ii)            Full form of VC (Vitamin C) needs to be given.

(iii)          pH determination needs to be mentioned wrt equipment used (with its make).

(iv)           Scientific names of microorganisms in antibacterial should be in italics.

4.     In results

(i)              Full form of OTUs be given while writing first time, many places OTU is written as OUT.

(ii)            The legends of all figures are incomplete which should be in a sentence and not as just abbreviations.

(iii)          At page-7, line 190,191, mitochondria and chloroplast have been mentioned as bacteria which is incorrect. The other genera (Pantoea and Gluconobacter) should be in Italics.

5.     It is suggested to combine discussion with results portion.

Reviewer 2 Report

Effect of bacterial community structure on the quality of sugar-cane juice during fermentation                                                                                                                                               canejuiceduringfermentation                                                                                                                                               2Cane juice during fermentation

Comments:

Line no 36: What are properties sugarcane have? Specify some of the properties so that it could be clear for the students to understand the role of sugarcane.

Line no 38: Describe the term blood tonifyining fruit so that it could become easy for readers to understand.

Line no 41: How sugar content of sugar cane juice could be measured and what is the score of measurement?

Line no 45: What are the threats faced by the consumers? Elaborate them so that it could be used to create awareness.

Line no 54:  Give examples of this study so that it could become easy to understand.

Line no 68: The results are not described so that it could be could be explained how it is related to this research.

Line no 79:  Chemicals which should be used in this study and analysis are not mentioned.

Line no 93: Purpose of different timings is not clear and it is a confusing statement.

Line no 102: Concentration of chemicals mentioned in this section is not written due to which it is difficult for the reader to explain the amount used.

Line no 125: Description of results is not described and values are not mentioned clearly.

Line no 138: Figure is not described which makes it a confusing scenario and it could not be understood.

Line no 150: In this section abbreviations are not described so it could not be understood by the reader.

Line no 172: . Figure is not described in this section so again it is a confusing statement

Line no 191: . Explain these difficult terms so that it could become easier for the reader to go through the whole document easily 

Line no 198: How this analysis is used in this study because it is not described.

Line no 238:  Why principal component analysis is used in this used in this study? Explain the purpose of this analysis?

Line no 265: In this section results are not defined clearly. Results should be mentioned clearly.

Line no 283: Why sugarcane has inhibitory effect on the bacteria. It should be define so that it becomes clearer

Line no 324: Which type of organic acids is produced by these pathogenic bacteria.

Line no 340: Why it is important to analyses pH in this section? What is the effect of pH?

Line no 357: What is the effect of production of lactic acid? Effects are not described.

Line no 364: In this study the results are not described and conclusion is not given which could relate it to the study

Overall the manuscript is not of high quality, and the requirements are required.

Author Response

Dear Reviewer,

Thank you for your comments on this article. Please see the attachment on reply.

Round 2

Reviewer 1 Report

The MS has been revised as per suggestions and error fixing. It seems fine for consideration for publication